# Pulmonary Embolism Associated with Olaparib in *BRCA2*-Mutated Prostate Cancer: A Case Report

**DOI:** 10.3390/curroncol32090523

**Published:** 2025-09-19

**Authors:** Shuhei Ishii, Shigekatsu Maekawa, Fumiko Amano, Daichi Kikuchi, Daiki Ikarashi, Renpei Kato, Mitsugu Kanehira, Ryo Takata, Jun Sugimura, Wataru Obara

**Affiliations:** 1Department of Urology, Iwate Medical University, 2-1-1 Idai-dori, Yahaba, Shiwa 028-3695, Iwate, Japan; sishii@iwate-med.ac.jp (S.I.); dikara@iwate-med.ac.jp (D.I.); rekato@iwate-med.ac.jp (R.K.); mkanehir@iwate-med.ac.jp (M.K.); rtakata@iwate-med.ac.jp (R.T.); jsugi@iwate-med.ac.jp (J.S.); watao@iwate-med.ac.jp (W.O.); 2Department of Urology, Iwate Prefectural Kamaishi Hospital, 10-483-6 Kasshi, Kamaishi 026-8550, Iwate, Japan; fmk9@hotmail.co.jp (F.A.); s15silvia_33skyline@yahoo.co.jp (D.K.)

**Keywords:** pulmonary embolism, olaparib, mCRPC, D-dimer, anticoagulation

## Abstract

Prostate cancer is common in men, and targeted medicines are improving survival. Olaparib is used when the cancer carries a harmful change in a gene that repairs genetic material. Evidence indicates that people with prostate cancer treated with olaparib experience serious clotting events in the veins and clots in the lungs more often than people with breast, ovarian, or pancreatic cancers. We report the case of a 70-year-old man with advanced prostate cancer who developed clots in the lungs seven months after starting olaparib. The diagnosis was confirmed by a contrast-enhanced chest scan and a blood test that reflects clot breakdown, and treatment with a blood-thinning medicine quickly improved symptoms and scan findings. To our knowledge, this is the first published case linking olaparib to clots in the lungs in prostate cancer. This report highlights the importance of careful symptom monitoring, simple risk assessment, and prompt testing during treatment. Early recognition and swift treatment may improve safety, outcomes, and shared decision-making when considering Olaparib.

## 1. Introduction

Olaparib, a poly (ADP-ribose) polymerase (PARP) inhibitor, has demonstrated significant clinical efficacy and is currently approved for the treatment of select patients with ovarian, breast, pancreatic, and prostate cancers. In patients with metastatic castration-resistant prostate cancer (mCRPC), particularly in those harboring mutations in *BRCA1*, *BRCA2*, or *ATM*, olaparib has been shown to markedly prolong radiographic progression-free survival and overall survival compared with next-generation androgen receptor pathway inhibitors such as enzalutamide or abiraterone [1].

Interestingly, among the currently approved PARP inhibitors, olaparib appears to be associated with the highest reported incidence of venous thromboembolism (VTE), particularly in populations with PC, compared with niraparib, rucaparib, and talazoparib, for which similar trends have not been observed in large-scale trials [2]. However, emerging evidence has raised concerns regarding the safety profile of olaparib, particularly with respect to thromboembolic events. In the PROfound trial, VTE developed in 7.8% of patients treated with olaparib compared with 3.1% in the control group, with pulmonary embolism (PE) reported in 4.7% and 0.8% of the individuals, respectively [3]. Notably, among the various malignancies treated with olaparib, PC appears to be associated with the highest incidences of both VTE (7.3%) and PE (6.5%) [1,4]. These rates markedly exceeded those associated with breast cancer (0%) [5,6], ovarian cancer (0–3.0%) [7,8,9], uterine cancer (1.3%) [7], and pancreatic cancer (0%) [8], suggesting a potential tumor type-specific thrombotic risk profile. Therefore, although the underlying mechanism is unknown, treatment with olaparib in patients with PC may affect blood coagulation.

Cancer-associated PE is a life-threatening complication that can markedly compromise clinical outcomes, particularly in patients with cancer [9]. Despite the recognition of an increased VTE risk associated with olaparib, there have been no published case reports specifically documenting olaparib-induced PE in patients with PC. Here, we report the first known case of PE during olaparib therapy in a patient with mCRPC harboring a deleterious *BRCA2* mutation. This case highlights the need for heightened clinical awareness and may provide new insights into the early recognition and management of olaparib-associated thromboembolic events.

## 2. Case Presentation

A 70-year-old man was diagnosed with PC (clinical stage cT1cN0M0). His prostate-specific antigen (PSA) level at diagnosis was 7.51 ng/mL, and prostate biopsy revealed acinar adenocarcinoma with a Gleason score of 4 + 3 = 7. The patient underwent intensity-modulated radiation therapy targeting the prostate (74 Gy/37 fractions). Two years later, multiple lymph node swellings and bone metastases were detected.

Combined androgen blockade (CAB) with degarelix and bicalutamide was initiated. However, PSA levels increased 11 months later, prompting a sequential change in treatment with flutamide, docetaxel, enzalutamide, and cabazitaxel. Palliative radiation was administered to spinal and iliac bone metastases (Figure 1). The patient underwent transurethral resection of the prostate for hemostasis and relief of urinary retention. Genomic profiling of transurethral resection of the prostate specimens by using FoundationOne^®^ CDx (Foundation Medicine, Cambridge, MA, USA) revealed a pathogenic *BRCA2* mutation, *BRCA2* c.9118-1G>A.

Olaparib therapy was initiated 55 months after initiating CAB, and it led to a marked decrease in PSA levels (Figure 1). Seven months after initiating olaparib (63 months from the initiation of CAB), the patient developed dyspnea and chest pain and was hospitalized. The patient’s clinical parameters were as follows: body temperature was 36.7 °C, blood pressure was 102/54 mmHg, pulse rate was 85 beats per minute, oxygen saturation was 91% with a nasal cannula delivering oxygen at 2 L/min, and serum D-dimer level was 45.3 µg/mL (normal: <1.0 µg/mL). The 12-lead electrocardiogram (ECG) did not reveal any ischemic changes, such as ST-T segment alterations. The Wells score was 4 points (Table 1), indicating an intermediate PE probability. Contrast-enhanced computed tomography (CT) revealed that although the lymph node swelling had decreased, there were multiple pulmonary emboli (Figure 2a), leading to a diagnosis of PE. Notably, aside from PE, there were no indications of venous thrombosis. Furthermore, there was no presence of fever and no elevation in white blood cell count, and CT excluded infectious conditions, including pneumonia.

The patient was administered oxygen and the direct oral anticoagulant apixaban (5 mg twice daily). CT performed after 15 days showed a marked reduction in all thrombi (Figure 2b), and D-dimer levels decreased (Figure 1). The patient’s respiratory distress was alleviated, and therefore, oxygen therapy was discontinued 35 days after the initiation of apixaban. Two months after olaparib cessation, PSA levels began to increase, although no metastases were observed on CT, bone scintigraphy, or fludeoxyglucose-positron emission tomography/CT. Therefore, abiraterone acetate (1000 mg/day) and prednisolone (10 mg/day) were started (67 months after CAB initiation); however, the PSA level continued to increase. Olaparib (half-dose) was restarted with the patient’s informed consent, because the thromboembolic event was assessed as grade 1 according to Common Terminology Criteria for Adverse Events (CTCAE) version 5.0, 70 months after CAB initiation. Nevertheless, PSA levels remained elevated, although there was no recurrence of PE. The patient eventually developed severe sepsis due to urinary tract infection and prostate hemorrhage and died 71 months after CAB initiation (2 months after the re-initiation of olaparib) (Figure 1).

## 3. Discussion

This report describes a case of PE in a patient with mCRPC undergoing olaparib therapy, highlighting a rare but clinically important adverse event that may affect treatment continuity and patient prognosis. Although olaparib effectively reduced PSA levels, its discontinuation owing to PE may have contributed to subsequent disease progression and mortality.

Cancer-associated thrombosis, encompassing both venous and arterial thromboembolic events, is a well-documented complication that adversely affects the prognosis and quality of life of patients with cancer [10]. VTE markedly worsens cancer outcomes; for example, the 1-year all-cause mortality rate in patients with VTE was 32.1%, compared with 11.0% in those without VTE (hazard ratio [HR]: 3.23, 95% confidence interval [CI]: 2.75–3.79) [11]. Although PC has traditionally been considered low risk for cancer-associated thrombosis [10,12], recent studies suggest that this risk increases substantially in the metastatic setting [13] and with certain treatments, including androgen deprivation therapy and newer targeted therapies, such as olaparib [14,15,16]. Mechanistically, thrombus formation in malignancies is driven by multiple factors including tumor-derived procoagulant activity, systemic inflammation, endothelial dysfunction, and treatment-induced hypercoagulability [17]. In our case, the patient had previously undergone multiple lines of systemic therapy and radiation for bone metastasis, each of which might have cumulatively contributed to the thrombotic risk. Notably, androgen deprivation therapy itself has been associated with a 22–84% increased risk of deep vein thrombosis and an 84–179% increased risk of PE compared with the corresponding risks in non-users [14,15,18,19]. Moreover, a population-based study had shown that patients with PC and VTE had a more than threefold increased risk of mortality (HR: 3.69, 95% CI: 3.55–3.83) compared with that in those without thromboembolism [16]. These data highlight the significance of PE as a potential turning point in the clinical trajectory of mCRPC. However, the precise pathophysiological mechanisms underlying olaparib-associated VTE remain unclear. PARP inhibition likely alters the DNA repair pathways in endothelial or immune cells, contributing to vascular injury or inflammation. Alternatively, the hypercoagulable state may reflect the synergistic effect of prior therapies and disease progression in genetically predisposed individuals. Regardless of the underlying mechanism, risk stratification and vigilant monitoring are warranted during olaparib treatment [20]. According to the appropriate use guide for olaparib in Japan, if a PE develops, treatment may be continued if it is of grade 1–2 according to CTCAE ver. 5.0. Temporary treatment discontinuation is recommended for PE of grade 3–4, and olaparib can be resumed without dose reduction once recovery to grade 1 is achieved. However, if the condition recurs after treatment resumption and is deemed unmanageable even after treatment has been discontinued again, dose reduction or discontinuation should be considered [21].

CT pulmonary angiography is the gold standard for PE diagnosis [22], and the D-dimer level remains a key biomarker in the diagnostic work-up of PE. In this case, the markedly elevated D-dimer level (45.3 µg/mL) at presentation and its subsequent decrease after anticoagulation mirrored the patient’s clinical course. Previous studies support the utility of D-dimer not only for diagnosis but also for monitoring of treatment response in patients with cancer-associated PE [23,24,25]. Additionally, the Wells score provides a useful estimate of the probability of PE and facilitates timely imaging and intervention [26,27]. Although the Khorana score is widely used to assess VTE risk in patients initiating chemotherapy, its predictive value diminishes over time and does not account for newer agents, such as PARP inhibitors [28]. A combined approach incorporating the Wells score and D-dimer level may facilitate early detection and management of thromboembolic events in high-risk patients (Table 1 and Table 2). Moreover, adenocarcinomas such as prostate cancer have been reported to induce systemic fibrinolytic activity through tumor-derived proteolytic enzymes, which can activate plasminogen and degrade fibrinogen. Although fibrinogen levels were not available in this case, such mechanisms may have contributed to the abrupt elevation of D-dimer in addition to the thrombotic event itself [29].

Direct oral anticoagulants have become the preferred treatment option over low-molecular-weight heparins because of their ease of administration and comparable efficacy. Among the direct oral anticoagulants, apixaban may offer a favorable bleeding risk profile, as demonstrated in the Caravaggio trial [30,31] and other comparative studies [32,33]. Furthermore, prophylactic use of apixaban has been shown to reduce VTE risk in ambulatory patients with cancer having elevated Khorana scores [28], although its utility in the context of olaparib therapy warrants further investigation.

## 4. Conclusions

This is the first reported diagnosis, treatment, and follow-up of PE associated with olaparib use in a patient with mCRPC. Although rare, PE is a potentially fatal complication that may be under-recognized in clinical practice. This report underscores the importance of early detection through clinical risk assessment and biomarker monitoring during PARP inhibitor therapy. Improving the clinical awareness of this potential adverse event may facilitate safer administration of olaparib and lead to improved outcomes in patients with mCRPC.

## Figures and Tables

**Figure 1 curroncol-32-00523-f001:**
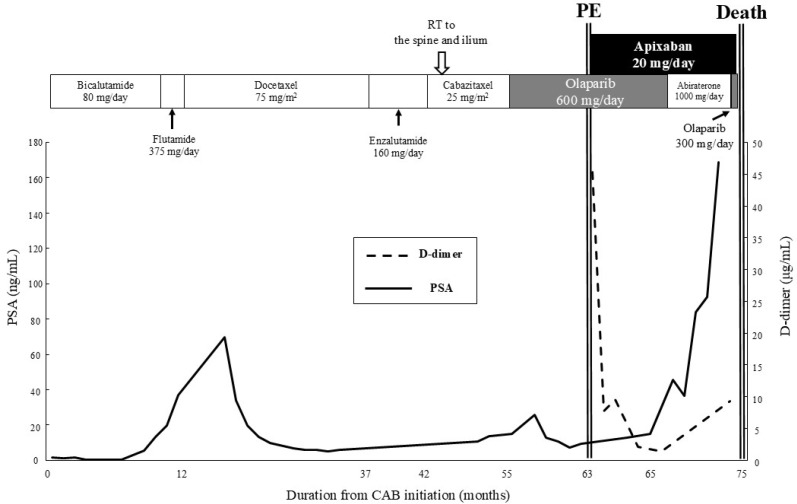
Treatment details, PSA before and after PE onset, and D-dimer levels. PSA: prostate-specific antigen; PE: pulmonary embolism.

**Figure 2 curroncol-32-00523-f002:**
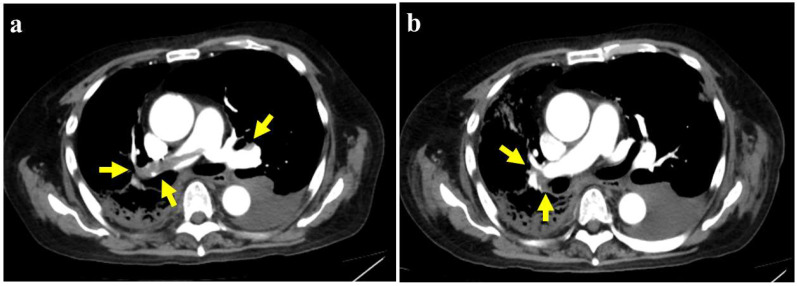
Contrast-enhanced computed tomography (**a**) Multiple emboli noted in the pulmonary arteries at the time of diagnosis (yellow arrows). (**b**) Fifteen days after the initiation of apixaban, a reduced number of emboli were observed in the pulmonary artery (yellow arrows).

**Table 1 curroncol-32-00523-t001:** Wells score.

Clinical Feature	Wells Score	Current Case
Clinical signs and symptoms of DVT	3	0
Pulmonary embolism as the most likely diagnosis	3	3
Heart rate of >100 beats per minute	1.5	0
Immobilization for at least 3 days or surgery in the preceding 4 weeks	1.5	0
Previous diagnosis of DVT or pulmonary embolism	1.5	0
Hemoptysis	1	0
Malignancy treatment within the preceding 6 months or palliative treatment	1	1
	Total score	4

Low risk: <2 points, Intermediate risk: 2–6 points, High risk: >6 points. PE unlikely: 0–4 points, PE likely: >4 points. DVT, deep vein thrombosis; PE, pulmonary embolism.

**Table 2 curroncol-32-00523-t002:** Khorana score.

		Khorana Score	Current Case
**Cancer type**	stomach	+2	0
	pancreas	+2	
	lung	+1	
	lymphoma	+1	
	gynecologic	+1	
	bladder	+1	
	testicular	+1	
	other	0	
**Pre-chemotherapy platelet count of ≥350 × 10^9^/L**	No: 0	Yes: +1	0
**Hemoglobin level of <10 g/dL or use of red blood cell growth factors**	No: 0	Yes: +1	0
**Pre-chemotherapy leukocyte count of >11 × 10^9^/L**	No: 0	Yes: +1	0
**BMI of ≥35 kg/m^2^**	No: 0	Yes: +1	0
**Total score**			0

Low risk: 0 points, intermediate risk: 1–2 points, high risk: >3 points.

## Data Availability

The original contributions presented in this study are included in the article. Further inquiries can be directed to the corresponding author.

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
