# Peer review of "Pulmonary Embolism Associated with Olaparib in BRCA2-Mutated Prostate Cancer: A Case Report"

_curroncol, 2025, doi:10.3390/curroncol32090523_

Round 1

Reviewer 1 Report

Comments and Suggestions for Authors

This case report documents a rare instance of pulmonary embolism (PE) associated with olaparib therapy in a patient with BRCA2-mutated metastatic castration-resistant prostate cancer. The authors provide a clear clinical timeline and highlight the need for vigilance regarding thromboembolic risks with PARP inhibitors. While the case adds to existing safety literature, its clinical impact is limited by the descriptive nature and absence of mechanistic or broader contextual analysis. Further research is needed to clarify the relationship between olaparib and PE. As detailed below some additional issues are required to consolidate claims

  1. While the authors claim this is the first reported case of olaparib-associated pulmonary embolism (PE) in metastatic castration-resistant prostate cancer (mCRPC), the report is largely descriptive and lacks depth in exploring the underlying mechanism or clinical implications. The occurrence of venous thromboembolism (VTE) with PARP inhibitors, including olaparib, has been described in prior clinical trials and pharmacovigilance reports. The value of this single case is limited unless it is integrated with a broader review or mechanistic insight
  2. The article does not provide a detailed description of the patient’s physical exam findings, such as presence of tachycardia, tachypnea, leg swelling, blood pressure
  3. The authors are advised to further explore or exclude other causes of the patient's symptoms (e.g., cardiac causes, infection).
Comments on the Quality of English Language

Need to improve

Author Response

We sincerely appreciate your thorough review of our manuscript and the insightful comments provided by the deputy editor. In accordance with these suggestions, we have carefully revised the manuscript, with the changes clearly highlighted in red within the text. We believe that these revisions have improved the clarity and completeness of our report, and we hope that the revised version addresses your concerns satisfactorily.

Comment 1: While the authors claim this is the first reported case of olaparib-associated pulmonary embolism (PE) in metastatic castration-resistant prostate cancer (mCRPC), the report is largely descriptive and lacks depth in exploring the underlying mechanism or clinical implications. The occurrence of venous thromboembolism (VTE) with PARP inhibitors, including olaparib, has been described in prior clinical trials and pharmacovigilance reports. The value of this single case is limited unless it is integrated with a broader review or mechanistic insight.

Response 1: Thank you very much for your valuable and constructive comments.

We fully agree that venous thromboembolism (VTE), including pulmonary embolism (PE), has been reported in clinical trials and pharmacovigilance data on PARP inhibitors, including olaparib. However, as we have highlighted in the revised manuscript, the incidence of VTE appears to be higher in prostate cancer compared to other malignancies such as breast or ovarian cancer, a fact that is not widely recognized among urologists. We believe this epidemiological difference deserves more attention in clinical practice.

Moreover, while the underlying mechanism of olaparib-induced VTE remains unclear, the reason why prostate cancer patients may be at greater risk has not been elucidated either. Possible explanations, such as the distinct tumor biology of prostate cancer, co-existing risk factors, or interactions with androgen receptor signaling inhibitors, remain speculative and warrant further investigation.

Therefore, we believe that documenting this case is meaningful for two reasons: (1) it represents, to our knowledge, the first reported case of PE occurring during olaparib treatment in metastatic castration-resistant prostate cancer, and (2) it provides detailed clinical information on the management strategy used, which may be of practical value for physicians who encounter similar situations.

We sincerely hope this clarification helps convey the significance of our report, and we appreciate your understanding of the rationale for its publication.

Comment 2: The article does not provide a detailed description of the patient’s physical exam findings, such as presence of tachycardia, tachypnea, leg swelling, blood pressure.

Response 2: Thank you very much for your constructive feedback. In response to your suggestion, we have revised the manuscript to provide a more detailed description of the patient’s physical examination findings and to clarify the exclusion of other possible causes.

Specifically, the text on page 2, line 82, has been replaced with the following sentence:

The patient’s clinical parameters were as follows: body temperature was 36.7°C, blood pressure was 102/54 mmHg, pulse rate was 85 beats per minute, respiratory rate was 18 breaths per minute, oxygen saturation was 91% with a nasal cannula delivering oxygen at 2 L/min, and serum D-dimer level was 45.3 µg/mL (normal: <1.0 µg/mL). Physical examination revealed no lower limb swelling, redness, or tenderness. The 12-lead electrocardiogram (ECG) did not show any ischemic changes, such as ST-T segment alterations.

Comment 3: The authors are advised to further explore or exclude other causes of the patient's symptoms (e.g., cardiac causes, infection).

Response 3: we added the following description to line 90:

Furthermore, there was no fever, no elevation in white blood cell count, and chest CT excluded infectious conditions, including pneumonia. Cardiac evaluation, including ECG and echocardiography, did not reveal findings suggestive of acute coronary syndrome or heart failure.

We believe these additions provide a clearer clinical picture and help demonstrate that alternative causes of the patient’s symptoms were carefully considered and excluded.

Reviewer 2 Report

Comments and Suggestions for Authors

This is an interesting case report on an patient with metastatic prostate cancer and pulmonary embolism under treatment with an ADP-ribose polymerase Olaparib. Although higher rates of thromboembolic complications have been described in treatment studies with Olaparib, a major limitation of a causal explanation in the present case report ist the underlying cancer. We can never know whether it was not the expanding tumor the trigger fort he thrombotic event. Further information on the case though could enrich discussion and promote understanding.

  • Authors might want to discuss that Adeno-Ca tumors, such as the prostate cancer, tend to induce a primary systemic fibrinolytic effect because tumor cells can produce proteolytic enzymes which activate plasminogen and/or lyse fibrinogen. The abrupt increase in D-Dimers might be such an effect in addition to the thrombotic event. Are there any information available on the fibrinogen levels at this stage?
  • Are there any information available on the localisation of the thrombotic event, was any distal or proximal deep vein thrombosis present, which might have ended-up in pulmonary embolism?
  • Authors might also want to discuss deeper the beneficial effect of oral anticoagulation as a tretament for tumor associated thrombosis and discuss the question whether Olaparib should be discontinued or not. Normally protection against thrombosis is present under normal dose of Apixaban, and medicaments which otherwise trigger thrombotic events could be further administered if useful, under anticoagulation protection (=still an expert opinion).

Author Response

We sincerely appreciate your thorough review of our manuscript and the insightful comments provided by the deputy editor. In accordance with these suggestions, we have carefully revised the manuscript, with the changes clearly highlighted in red within the text. We believe that these revisions have improved the clarity and completeness of our report, and we hope that the revised version addresses your concerns satisfactorily.

Comment 1: Authors might want to discuss that Adeno-Ca tumors, such as the prostate cancer, tend to induce a primary systemic fibrinolytic effect because tumor cells can produce proteolytic enzymes which activate plasminogen and/or lyse fibrinogen. The abrupt increase in D-Dimers might be such an effect in addition to the thrombotic event. Are there any information available on the fibrinogen levels at this stage?

Response 1: Thank you very much for this insightful and educational comment. Unfortunately, fibrinogen levels were not measured in this case, and therefore we are unable to provide direct data on this parameter. Nevertheless, we fully acknowledge the importance of your suggestion. Adenocarcinomas, including prostate cancer, have been reported to induce systemic fibrinolytic activity through the production of proteolytic enzymes capable of activating plasminogen and degrading fibrinogen. As you correctly pointed out, such mechanisms could potentially contribute to the abrupt elevation in D-dimer levels observed in our patient, in addition to the thrombotic event itself.

In response to your valuable advice, we have revised the discussion to reflect this perspective, which we believe adds depth and broader clinical context to the interpretation of our findings, even though specific fibrinogen measurements were not available in this case.

Specifically, we added the following sentence on page 6, line 176 of the revised manuscript:

Moreover, adenocarcinomas such as prostate cancer have been reported to induce systemic fibrinolytic activity through tumor-derived proteolytic enzymes, which can activate plasminogen and degrade fibrinogen. Although fibrinogen levels were not available in this case, such mechanisms may have contributed to the abrupt elevation of D-dimer in addition to the thrombotic event itself.”

Comment 2: Are there any information available on the localisation of the thrombotic event, was any distal or proximal deep vein thrombosis present, which might have ended-up in pulmonary embolism?

Response 2: Thank you very much for this valuable comment. In order to clarify the localization of the thrombotic event, we have revised the manuscript accordingly. Specifically, on page 2, line 89, we added the following sentence:

Notably, aside from PE, there were no indications of venous thrombosis.

This addition emphasizes that no evidence of either distal or proximal deep vein thrombosis was observed in this case, thereby supporting the interpretation that the pulmonary embolism developed without detectable peripheral venous thrombosis. We believe that this clarification improves the completeness of the clinical description and directly addresses the reviewer’s concern.

Comment 3: Authors might also want to discuss deeper the beneficial effect of oral anticoagulation as a treatment for tumor associated thrombosis and discuss the question whether Olaparib should be discontinued or not. Normally protection against thrombosis is present under normal dose of Apixaban, and medicaments which otherwise trigger thrombotic events could be further administered if useful, under anticoagulation protection (=still an expert opinion).

Response 3: Thank you very much for this thoughtful and important comment. We agree that the management of tumor-associated thrombosis in patients receiving olaparib deserves careful discussion, particularly with regard to the role of oral anticoagulation and the decision of whether or not to discontinue olaparib.

As reported in the PROfound trial, among 17 patients in the olaparib group who developed venous thromboembolism (VTE), treatment discontinuation occurred in only 2 cases, and temporary interruption in 3 cases. This suggests that, as you correctly noted, continuation of olaparib under anticoagulation might be a reasonable option in certain circumstances. However, it is also noteworthy that 8 of these 17 cases (47.1%) experienced Grade ≥3 VTE. Given this substantial proportion of severe events, we considered temporary discontinuation of olaparib in our patient to be a prudent and safety-oriented approach.

We have now incorporated this discussion into the revised manuscript, emphasizing that while oral anticoagulation (e.g., apixaban) may provide protection and allow continuation of therapy in some cases, the absence of prior case reports in prostate cancer and the potential for severe thrombotic complications justified our conservative management strategy in this instance. We believe this addition strengthens the clinical relevance of our report and addresses the important point you raised.